# Use of Selective Serotonin Reuptake Inhibitors Is Associated with a Lower Risk of Colorectal Cancer among People with Family History

**DOI:** 10.3390/cancers14235905

**Published:** 2022-11-29

**Authors:** Naiqi Zhang, Jan Sundquist, Kristina Sundquist, Jianguang Ji

**Affiliations:** 1Center for Primary Health Care Research, Lund University, Region Skåne, 205 02 Malmö, Sweden; 2Department of Family Medicine and Community Health, Department of Population Health Science and Policy, Icahn School of Medicine at Mount Sinai, New York, NY 10029-6574, USA; 3Center for Community-Based Healthcare Research and Education (CoHRE), Department of Functional Pathology, School of Medicine, Shimane University, Shimane 693-8501, Japan

**Keywords:** selective serotonin reuptake inhibitors, colorectal cancer, family history, chemoprevention

## Abstract

**Simple Summary:**

Individuals with a family history of colorectal cancer (CRC) are at a high risk of developing CRC. We performed a nationwide cohort study to explore whether selective serotonin reuptake inhibitors (SSRIs) play a role in preventing CRC among individuals with family history. This report suggests that SSRI use was associated with a reduced incidence of CRC among people with a family history of CRC, and the decreased risk of CRC showed a non-linear, dose-dependent pattern. In addition, we found that the use of SSRIs was strongly associated with CRC diagnosed at an advanced stage rather than at an earlier stage, suggesting that the SSRIs might reduce the metastatic potential of cancer cells leading to stage migration. This population-based cohort study suggests that the use of SSRIs is associated with a reduced risk of CRC among individuals with a family history of CRC.

**Abstract:**

Individuals with a family history of colorectal cancer (CRC) are at a high risk of developing CRC. Preclinical and population-based evidence suggests that selective serotonin reuptake inhibitors (SSRIs) might play a role in preventing CRC. We performed a nationwide cohort study to explore whether the use of SSRIs could reduce CRC risk among individuals with family history. We identified individuals aged 50 and above who had one or more first-degree relatives diagnosed with CRC. A total of 38,617 incident SSRI users were identified and matched with 115,851 non-users, on a ratio of 1:3. The Cox regression model was used to calculate hazard ratios (HRs) and 95% CI confidence intervals (CIs). We found a significant negative association between SSRI use and the risk of CRC (adjusted HR, 0.77; 95% CI, 0.70–0.85). Restricted cubic spline regression showed a non-linear dose-responded relationship between SSRI use and CRC risk. The association was stronger in rectal cancer than colon cancer (adjusted HR, 0.73 vs. 0.79), and more pronounced in advanced-stage CRC than early-stage CRC (adjusted HR, 0.73 vs. 0.80). This population-based cohort study suggests that the use of SSRIs is associated with a reduced risk of CRC among individuals with a family history of CRC.

## 1. Introduction

With an estimated 1.9 million new cases and 1 million deaths across the globe in 2020, colorectal cancer (CRC) continues to be ranked as the third most common cancer and the second leading cancer death worldwide [1]. The causative mechanisms of cancer are not completely and comprehensively under-established. Around 70–80% of CRC cases are assumed to be driven by environmental factors, which include social, cultural, and lifestyle factors [2], while the rest is deemed to be triggered by the genetic factors of inheritance [3]. Many epidemiological studies have shown that people who have a first-degree relative diagnosed with CRC have about two times the lifetime risk of developing CRC [4], possibly because they have similar genetic backgrounds along with environmental factors. Given the fact that the risk of developing CRC is at a relatively high level, an effective prevention strategy to reduce the risk in individuals with a family history is urgently needed.

Chemoprevention is an appealing cancer prevention strategy which uses pharmaceutic agents to hinder the pathways that cause disease; such an approach has been successfully applied to CRC [5]. Non-cancer drugs, including selective serotonin reuptake inhibitors (SSRIs), are proven to have potential beneficial effects in protecting against CRC [6]. SSRIs are broadly prescribed to treat a diverse range of diseases, such as depression, anxiety, insomnia, and chronic pain. It increases the level of serotonin in the brain by blocking neural reabsorption. Having more serotonin in the synapses implies that it can pass on information with greater ease and all known SSRIs appear to work in this way. Previous evidence from population-based epidemiological studies has implied that SSRIs may have the potential to lower the risk of many cancers, including kidney, ovarian, liver, and colorectal cancers [7,8,9,10]. A number of studies support the role of SSRIs against CRC, and recent in vitro and in vivo studies highlight the anti-tumor effect of SSRIs, suggesting that SSRIs may be the feasibly chemopreventive agents for the onset of CRC [11,12]. However, a meta-analysis comprising six observational studies reported an overall risk as marginally insignificant (adjusted OR 0.89, 95% CI 0.79–1.01) [13]. Most of the previously conducted studies exploring the anti-cancer effects of SSRIs have mainly addressed the general population, whereas individuals at a high risk, such as those with a family history of CRC, are urgently required to be evaluated if SSRIs are high efficiency, chemopreventive drugs. There are currently five SSRIs prescribed in Sweden: fluoxetine, citalopram, paroxetine, sertraline, and escitalopram. By integrating several national registries in Sweden, we aimed at exploiting the link between exposure to SSRIs and CRC risk in people with a family history of CRC.

## 2. Materials and Methods

### 2.1. Data Sources

This nationwide, retrospective cohort study was approved by the Ethics Committee of Lund University, Sweden, on 6 February 2013 (Dnr 2012/795). By linking to the Swedish Multi-generation Register and the Swedish Cancer Registry, we identified all individuals aged 50 and older who had one or more first-degree relatives (parents or siblings) diagnosed with CRC by using the 10th International Classification of Diseases codes C18, C19, and C20 (*n* = 285,008). There are more than 12 million entries in the Swedish Multi-generation Registry, which has been used for many analyses about the familial risk of cancer. Nearly 100% of the entire Swedish population is covered by the Swedish Cancer Registry, which is based on mandatory reports from clinical doctors and pathologists/cytologists.

### 2.2. Assessment of Selective Serotonin Reuptake Inhibitors Use

We extracted information from the Swedish Prescription Drug Register concerning the prescription of SSRIs in individuals with a family history of CRC using the Anatomical Therapeutic Chemical Classification (ATC) code N06AB. To each subtype of SSRIs, a seven-digit ATC code could be found: fluoxetine (N06AB03), citalopram (N06AB04), paroxetine (N06AB05), sertraline (N06AB06), and escitalopram (N06AB10). The Swedish Prescription Register contains all dispensed drug information serving the entire Swedish population with no more than 0.3% missing data since its creation on 1 July 2005 [14]. Dispensation records contain the date of dispensation, ATC code, and defined daily dose (DDD), which is defined as the assumed average daily dose for the primary indication in adults. We adopted a new-user study design with a six-month washout period to exclude prevalent SSRI users. The entry date was set to 1 January 2006, thus excluding individuals who were issued SSRIs between July 2005 and December 2005 (*n* = 23,118).

SSRI users were randomly assigned three comparators who did not receive prescriptions for SSRIs and did not experience CRC prior to the corresponding individual’s first SSRI prescription date (index date) based on age and sex.

### 2.3. Assessment of Outcome

Based on the Swedish Cancer Registry, we identified patients diagnosed with colon or rectal cancer between 1 January 2006, and 31 December 2018, with ICD codes C18, C19, and C20. TNM staging data are available in the Swedish Cancer Registry, including tumor size (T), nodal status (N), and metastatic disease (M). CRC stages can be determined by combining the T, N, and M categories, ranging from stage I to stage IV (most advanced), based on their combination as follows: stage I (T1 or T2), stage II (T3 or T4), stage III (any T, N1 or N2, M0), and stage IV (any M1). In addition, we were able to identify individuals who died during the follow-up phase by connecting to the Cause of Death Register.

Participants were followed up from January 2006 to (i) the date of the diagnosis of CRC, (ii) the date of death, or (iii) the end of this study (31 December 2018). This study excluded patients with a follow-up period of fewer than three months to minimize reverse causality.

### 2.4. Assessment of Covariates

Based on data retrieved from the National Patient Register and Statistics Sweden’s Total Population Register, several potential confounding factors were determined, including age (continuous variable), country of birth (Sweden or other), sex (male or female), years of education (1–9, 10–11, ≥12 years), income (lowest, middle-low, middle-high, and high), history of inflammatory bowel disease (IBD, including Crohn’s disease or ulcerative colitis, yes/no), history of colonoscopy within the last 10 years (yes/no), frequency of outpatient visits (0, 1, 2, ≥3 times/per year), obesity (identified from the National Patient Register using ICD-10 code “E66”, yes/no), chronic obstructive pulmonary disease (COPD, yes/no) as a proxy for smoking, prescription of other medication (aspirin, metformin, and statin, yes/no), and Charlson Comorbidity Index score (CCI, 0, 1, 2, ≥3). 

A personally identifiable number in Sweden synchronized the different registers, then overrides them with a serial number to ensure anonymity.

### 2.5. Statistical Analysis

We used Cox-regression models to calculate hazard ratios (HRs) and 95% confidence intervals (CIs) for CRC associated with SSRI use. Considering that the study population was the elderly with high mortality rates [15] and the suicide rate in SSRI users was higher than the general population [16], we computed death as a competing event and control the competing risk of death [17]. Final multivariate models were adjusted for age, sex, country of birth, highest education level, history of IBD, colonoscopy, obesity, COPD, outpatient visit frequency, use of aspirin, statin, and metformin. In addition, we calculated cumulative defined daily doses (cDDDs) of SSRIs, the sum of all prescribed DDDs during the follow-up period. We then performed a dose-response analysis using restricted cubic spline (RCS) curves based on a multivariate Cox proportional hazards model. To determine the best-fit curve for SSRI dose and CRC incidence, we tested three to seven cutoff points for RCS and selected the model with the lowest Akaike information criterion (AIC). Five cutoffs (0.05, 0.275, 0.5, 0.725, and 0.95) were used in the final model. In addition, we examined the relationship between SSRI use and the cancer site (colon, rectum) as well as cancer stage (stage I or II, stage III or IV). Stratified analyses by age and sex were also carried out to estimate the interactive effects of SSRIs on CRC risk.

We conducted some sensitivity analyses with the aim to explore the chance of coincidences in the results. First, we performed a sensitivity analysis to lag the first prescription of SSRIs by one year in order to avoid reverse causality and to take into account the biological latency period. Second, we analyzed death and diagnosis of other cancers as competing events to further control the competing risks of other cancers. Third, there was an exclusion of individuals who had been diagnosed with benign colorectal tumors from the study. Fourth, we assessed possible indication bias associated with SSRI use by investigating the relationship between tricyclic antidepressant (TCA) use and CRC risk. In Sweden, TCA is also an extensively prescribed agent for the management of depression. Fifth, to further assess whether our results were subject to an indication bias in depression, we performed a Mendelian randomization analysis to investigate the association between depression and CRC. Single nucleotide polymorphisms as genetic instrument variables were derived from pooled data from a GWAS meta-analysis of depression that included seven cohorts [18]. The GWAS summary data for CRC were collected from the UK biobank [19]. A couple of MR methods were carried out to be used in determining the estimates of depression on CRC, respectively MR-Egger, weighted median, and inverse variance weighting, after coordinating the effect alleles of these two GWAS.

R (package Two Sample MR) was used for the Mendelian randomized analysis. All other analyses were performed using SAS, version 9.4 (SAS Institute, Cary, NC, USA). 

## 3. Results 

The flowchart for the study design can be seen in Figure 1. 

A total of 154,468 individuals with at least one first-degree relative diagnosed with CRC were recruited for this study. SSRI users and comparisons were matched with similar distributions of age and sex. Compared with non-users, SSRI users had a slightly lower proportion of highest education level and highest income, a higher proportion of inflammatory bowel disease, obesity, COPD, colonoscopy screening, aspirin, metformin, and statin use, and higher outpatient visits and CCI scores (Table 1). The final multivariable regression model adjusted all the variables listed in Table 1.

Following a mean follow-up of 6.8 years, the incidence of CRC among SSRI users was 1.65/1000 person-years, which was significantly lower than among comparators not taking SSRIs (2.00/1000 person-years). In individuals with a family history, there was an inverse association between SSRI use and CRC risk; the crude HR was 0.77 (95% CI, 0.70–0.85) and the adjusted HR was 0.77 (95% CI, 0.70–0.85) (Table 2). We also investigated the effect of different SSRI types. There was a significant reduction in CRC risk for patients taking citalopram and sertraline, but not for those taking fluoxetine, paroxetine, or escitalopram, likely due to the very limited number of cases. We observed a non-linear dose-response relationship between SSRI use and CRC risk. As shown in Figure 2, a decrease in CRC incidence was found as SSRI doses were increased. The trend of the dose–response correlation was significant (*p* < 0.001). The inverse relationship was observed slightly stronger for rectal cancer (0.73; 95% CI, 0.63–0.91) when compared with colon cancer (0.79; 95% CI, 0.70–0.89), and slightly stronger for advanced stage (adjusted HR, 0.73; 95% CI, 0.63–0.85) than for early stage (adjusted HR, 0.80; 95% CI, 0.68–0.93).

Findings from stratified analyses are listed in Table 3. The association between SSRI use and CRC risk was significant among adults aged 60 to 69 years (adjusted HR, 0.61, 95% CI, 0.51–73) but not among adults aged 50 to 59 years (adjusted HR, 0.91; 95% CI, 0.76–1.09) and adults aged 70 years or older (adjusted HR, 0.86; 95% CI, 0.73–1.01). As stratified by gender, women may benefit more from using SSRIs (adjusted HR, 0.74; 95% CI, 0.64–0.84) when compared with men (adjusted HR, 0.84; 95% CI, 0.72–0.97).

In Table 4, we list the results of sensitivity analyses. In sensitivity analysis 1, the use of SSRIs continued to be associated with a reduced risk of CRC (adjusted HR, 0.82; 95% CI, 0.73–0.93) after lagging for one year after the initiation of SSRIs. In sensitivity analysis 2, the result remained robust with the main analysis when death and cancer diagnosis were both included as competing events (adjusted HR, 0.76; 95% CI, 0.69–0.84). In sensitivity analysis 3, SSRI use continued to be associated with a reduced risk of CRC (adjusted HR, 0.77; 95% CI, 0.70–0.85) after excluding patients with diagnoses of benign colorectal tumors. In sensitivity analysis 4, TCA use was not associated with CRC risk among individuals with a family history of CRC, with an adjusted HR of 1.02 (95% CI, 0.76–1.39). In sensitivity analysis 5, The results of Mendelian randomized analyses using different approaches (MR-Egger, weighted median, and inverse variance weighted) all revealed no significant association between depression and CRC risk (see more details in Table 5).

## 4. Discussion

The present cohort study, with the data from nationwide registers in Sweden, showed that SSRI use was associated with a reduced incidence of CRC among people with a family history of CRC, and the decreased risk of CRC showed a non-linear dose-dependent pattern. In addition, we found that the use of SSRIs was strongly associated with CRC diagnosed at an advanced stage rather than at an earlier stage, suggesting that the SSRIs might reduce the metastatic potential of cancer cells that lead to stage migration. 

People with a family history of CRC are at a higher risk of developing the disease. Approximately 30% of CRC patients have one or more relatives diagnosed with CRC [20]. Results from systematic reviews consistently show that the risk of developing CRC is twice as high in people with a family history of CRC as in the general population, and the risk increases with the number of affected relatives [21] as well as the younger age of CRC diagnosis [22]. However, there is limited epidemiological evidence for chemoprevention against CRC, especially when focusing on the high-risk group. The results from the present study suggest that SSRIs can protect against the development of CRC, and the effect is even stronger in patients diagnosed at an advanced stage. Nevertheless, there was no defined population-based evidence indicating the role of SSRIs in protecting against CRC up until now. A case-control study observed that regular use of SSRIs was significantly associated with a 45% decreased risk of developing CRC [7]. A Canadian registry-based nested case-control study also reported a 30% reduced risk among individuals with high cumulative doses of SSRIs before diagnosis (0–5 years) [23]. However, studies from Denmark, Finland, and the US showed little evidence of a protective association between SSRI use and CRC risk [24,25,26,27,28]. Additionally, cancer patients are often prescribed SSRIs to treat depression since cancer triggers depression [29]. SSRI use could improve the overall quality of life in advanced cancer patients [30], decrease the time of disease progression among ovarian cancer patients [31], and markedly improve cancer-specific survival in kidney cancer patients [32]. Hereafter, cancer treatment can be assisted by SSRIs.

While the biological mechanisms of the anti-tumor effect of SSRIs are not fully understood, some hypotheses have been proposed and validated in vivo or in vitro showing that SSRIs implement anti-cancer effects in a variety of ways. First, it has been suggested that SSRIs may stimulate the activation of serotonin receptors 5-HT1A, 5-HT2A, and 5-HT7A to boost the downstream MAPK/ERK pathway [33], which is highly involved in the processes of apoptosis, gene transcription, cell division, and cell cycle arrest [34]. Second, fluoxetine and sertraline could inhibit the proliferation of colon cancer cells by activating the JNK-c-Jun pathway [35,36]. Third, fluoxetine and escitalopram were also reported to inhibit the expression of proteins involved in the DNA repair mechanism and to suppress the NF-κB signaling pathway to reduce the cancer cell metastatic potential [37,38]. Fourth, fluoxetine and sertraline could elevate the phosphorylation of AMPK and block downstream mTOR pathway, resulting in autophagic cell death in various cancer cells [39,40,41], and enhance the sensitivity of several chemotherapy drugs on colon cancer cells [42]. In addition, results from animal models suggested that fluoxetine acts against the development of malignant microvessels and the expression of vascular endothelial growth factor (VEGF) and cyclooxygenase-2 (COX-2). It might control the carcinogenic interaction among microenvironment and modulate serotonin metabolism activity that leads to oncostatic effects on carcinogenic colon tissue [43]. Furthermore, SSRIs are likely to have an effect on cancer stem cells. In vitro experiments showed that SSRIs stimulated apoptosis and autophagy in glioma stem cells. Since the presence of cancer stem cells results in metastasis and invasion, along with resistance to chemotherapy, it is promising for the application of SSRIs as specific therapies targeting the serotonin system in cancer stem cells [44]. Furthermore, SSRIs might be a potential agent to treat the adverse effect of chemotherapy. Oxaliplatin is a commonly used platinum-based chemotherapy drug for colorectal cancer. SSRI was reported to inhibit oxaliplatin-induced peripheral neurotoxicity in a dose-dependently way [45].

The major strength of this work is that the study was performed based on the population at a national level. The cohort study design and large sample size ensured statistical power and minimized reversal causality. Data collected from nationwide registers could efficiently eliminate recall bias and minimize selection bias. Furthermore, register-based data provided us with information on potential confounding factors based on demographics and clinical characteristics. The design also allowed the assessment of the dose-response relationship between SSRI exposure and CRC. However, several limitations should be taken into account. First, due to individuals who used SSRIs having higher chances of visiting their doctors and caring more about their health, the results might be skewed by detection bias. However, we adjusted the regression model to account for colonoscopy, which is offered in Sweden in an opportunistic manner; thus, it can be used as a proxy of health behavior since people with greater health awareness and knowledge are more likely to undergo colonoscopies. Furthermore, outpatient visits were adjusted since they are strongly related to an adherence of CRC screenings. Using colonoscopy and the frequency of outpatient visits as a proxy of healthcare engagement might partly exclude the contribution of different health behavior among individuals who use SSRIs and those who do not use them. Our stratification analyses showed that the use of SSRIs was strongly associated with CRC diagnosed at an advanced stage rather than at an earlier stage, suggesting such detection bias might play a small role in our observation. Second, our results might be influenced by indication bias. To rule out the possibility of potential confounding by indication bias, we conducted a couple of sensitivity analyses, such as an active comparator study design that investigated the subsequent risk of CRC among individuals who have used TCA based on the fact that TCA is also a widely prescribed medication to treat depression. Additionally, we conducted a Mendelian randomized analysis to investigate the association between depression (the main indication for SSRIs) and CRC and did not find a significant relationship, suggesting that the observed results from our study might not be confounded by indication bias. Third, smoking, alcohol consumption, and dietary factors, which might confound our results, are not available in our nationwide databases. However, our regression models include COPD as a proxy for smoking. Although it is a crude proxy for smoking, it might partly exclude the confounding effect of smoking. Education status and income were also adjusted, which were strongly associated with lifestyle factors and might partially compensate for their confounding effects. Fourth, as the study had a relatively short follow-up time, it was not possible to examine the anticancer effect in young adults due to the small number of cases. Future studies utilizing larger sample sizes and longer follow-ups could explore this subject further. Fifth, our study used the administered data without detailed information regarding histological subtypes, such as information about microsatellite instability; thus, we could not explore the heterogeneity regarding the association between SSRIs and CRC by its microsatellite instability. Further studies are highly needed to explore this issue to provide a mechanistic interpretation of our findings. 

## 5. Conclusions

In summary, this population-based cohort study suggests that the use of SSRIs was associated with a decreased risk of CRC among people with a family history of CRC, and the decrease showed a dose-dependent pattern. Our results suggest a potential for SSRIs to be applied as chemoprevention against CRC in the high-risk group. Findings from this study need to be confirmed by well-designed randomization clinical studies in the future to provide a strong level of clinical evidence.

## Figures and Tables

**Figure 1 cancers-14-05905-f001:**
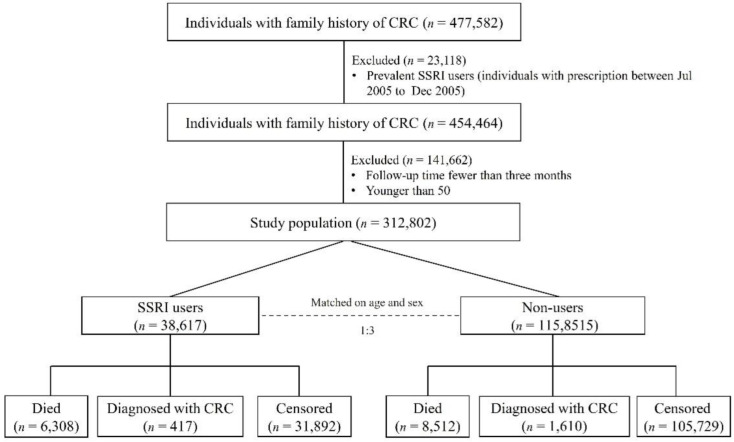
Flow chart of participants involved in this national cohort study.

**Figure 2 cancers-14-05905-f002:**
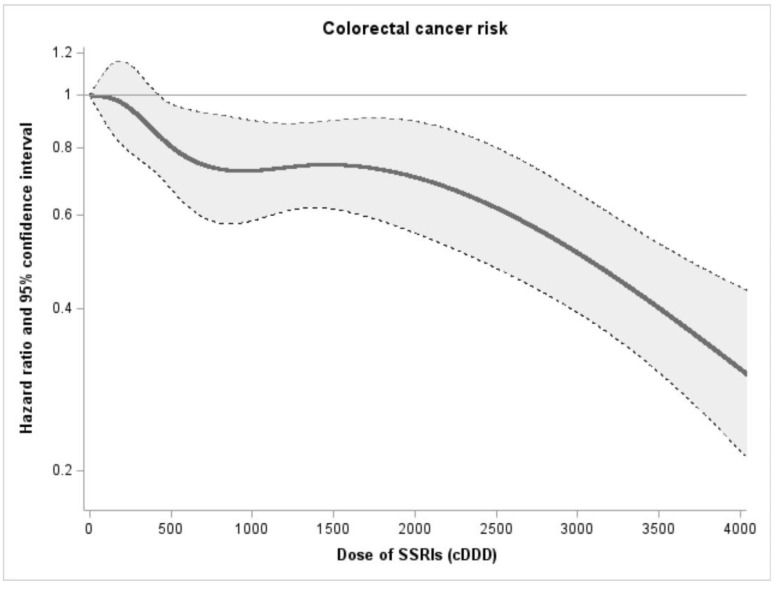
Association between different doses of SSRI use and the risk of colorectal cancer using restricted cubic spline, with 95% confidence intervals (the gray area). Adjusted for age at index, sex, education, birth country, income, history of inflammatory bowel disease, COPD, obesity, outpatient visits, history of colonoscopy, use of aspirin, use of statin, use of metformin, CCI.

**Table 1 cancers-14-05905-t001:** Demographic and clinical characteristics among SSRIs users and matched controls in people with a family history of CRC.

	SSRIs Users (*n* = 38,617)		Non-Users (*n* = 115,851)	
	No.	%	No.	%
Age at index				
50–59	14,584	37.8	43,752	37.8
60–69	13,275	32.4	39,825	32.4
≥ 670	10,758	27.8	32,274	27.8
Sex				
Males	15,678	40.6	47,034	40.6
Females	22,939	59.4	68,817	59.4
Birth country				
Sweden	37,788	97.8	113,389	97.8
Others	829	2.2	2462	2.2
Highest education level, year				
1–9	9931	25.7	29,432	25.4
10–11	16,735	43.3	49,790	43.0
≥12	11,951	31.0	36,629	31.6
Income				
Lowest	7608	19.7	22,217	19.2
Middle-low	10,058	26.1	29,929	25.8
Middle-high	9577	24.8	27,139	23.4
Highest	11,374	29.4	36,566	31.6
Inflammatory bowel disease				
No	37,985	98.4	114,482	98.8
Yes	632	1.6	1369	1.2
Obesity				
No	37,703	97.6	114,430	98.8
Yes	914	2.4	1421	1.2
COPD				
No	34,925	90.4	108,899	94.0
Yes	3692	9.6	6952	6.0
Colonoscopy				
No	35,811	92.7	110,623	95.5
Yes	2806	7.3	5228	4.5
CCI				
0	25,667	66.5	91,290	78.8
1	7873	20.4	16,738	14.4
2	2809	7.3	5080	4.4
≥3	2268	5.8	2743	2.4
Outpatient visits, per year				
0	13,792	35.7	54,163	46.7
1	14,085	36.5	33,925	29.3
2	5633	14.6	12,723	11.0
≥3	5107	13.2	15,040	13.0
Prescription of other medicines				
Aspirin	8065	20.9	16,113	13.9
Metformin	2616	6.8	5872	5.1
Statin	10,700	27.7	23,667	20.4

**Table 2 cancers-14-05905-t002:** Hazard ratios and 95% confidence intervals of colorectal cancer associated with SSRI use among individuals with a family history of CRC.

	Individuals, *n*	Person-Years	Cancer Diagnoses, *n*	IR, Per 1000 Person-Year	Crude			Adjusted *		
					HR	95% CI	*p* Value	HR	95% CI	*p* Value
SSRIs use										
Non-users	11,5851	804,265	1610	2.00	1			1		
SSRIs users	38,617	253,019	417	1.65	0.77	0.70–0.85	<0.001	0.77	0.70–0.85	<0.001
Subtype of SSRIs										
Fluoxetine	1515	14,224	21	1.47	0.74	0.49–1.13	0.164	0.75	0.47–1.18	0.210
Citalopram	15,587	129,042	227	1.76	0.77	0.68–0.88	<0.001	0.77	0.67–0.88	<0.001
Paroxetine	1116	10,759	13	1.21	0.64	0.38–1.09	0.103	0.66	0.36–1.20	0.172
Sertraline	9761	81,764	117	1.43	0.71	0.60–0.85	<0.001	0.69	0.57–0.83	<0.001
Escitalopram	3995	34,586	51	1.47	0.77	0.58–1.01	0.055	0.84	0.64–1.11	0.219
Cancer site										
Colon cancer										
Non-users	11,5851	804,265	1125	1.40	1			1		
SSRIs users	38,617	253,019	298	1.18	0.79	0.71–0.89	<0.001	0.79	0.70–0.89	<0.001
Rectal cancer										
Non-users	115,851	804,265	485	0.6	1			1		
SSRIs users	38,617	253,019	119	0.47	0.73	0.61–0.87	<0.001	0.73	0.63–0.91	0.003
Cancer stage										
Stage I and II										
Non-users	115,851	804,265	616	0.77	1			1		
SSRIs users	38,617	253,019	169	0.67	0.82	0.72–0.95	0.010	0.80	0.68–0.93	0.004
Stage III and IV										
Non-users	115,851	804,265	785	0.98	1			1		
SSRIs users	38,617	253,019	186	0.74	0.71	0.61–0.82	<0.001	0.73	0.63–0.85	<0.001

* Adjusted for age at index, sex, education, birth country, family history of CRC, history of inflammatory bowel disease, COPD, obesity, history of colonoscopy, use of aspirin, use of statin, use of metformin, CCI.

**Table 3 cancers-14-05905-t003:** Hazard ratios and 95% confidence intervals of different types of colorectal cancer associated with SSRI use among individuals with a family history of CRC, stratified by age and sex.

	Individuals, *n*	Person-Years	Cancer Diagnoses, *n*	IR, Per 1000 Person-Year	Crude			Adjusted *		
					HR	95% CI	*p* Value	HR	95% CI	*p* Value
Age										
50–59										
Non-users	43,752	339,128	392	1.16	1			1		
SSRIs users	14,584	110,460	121	1.1	0.92	0.77–1.10	0.353	0.91	0.76–1.09	0.303
60–69										
Non-users	39,825	296,312	657	2.22	1			1		
SSRIs users	13,275	92,807	134	1.44	0.61	0.52–0.73	<0.001	0.61	0.51–0.73	<0.001
≥70										
Non-users	32,274	168,826	561	3.32	1			1		
SSRIs users	10,758	49,752	162	3.26	0.86	0.74–0.99	0.048	0.86	0.73–1.01	0.064
Sex										
Male										
Non-users	47,034	316,661	696	2.2	1			1		
SSRIs users	15,678	97,325	193	1.98	0.83	0.72–0.95	0.008	0.84	0.72–0.97	0.018
Female										
Non-users	68,817	487,605	914	1.87	1			1		
SSRIs users	22,939	155,694	224	1.44	0.73	0.64–0.83	<0.001	0.74	0.64–0.84	<0.001

* Adjusted for age at index, sex, education, birth country, income, history of inflammatory bowel disease, COPD, obesity, outpatient visits, history of colonoscopy, use of aspirin, use of statin, use of metformin, CCI.

**Table 4 cancers-14-05905-t004:** Sensitivity analyses.

	Individuals, *n*	Person-Years	Cancer Diagnoses, *n*	IR, Per 1000 Person-Year	Crude			Adjusted *		
					HR	95% CI	*p* Value	HR	95% CI	*p* Value
Sensitivity analysis 1 ^†^										
Non-users	110,506	800,862	1458	1.82	1			1		
SSRIs users	36,138	251,473	374	1.49	0.82	0.73–0.93	0.001	0.82	0.73–0.93	0.001
Sensitivity analysis 2 ^‡^										
SSRIs users	113,007	784,061	1570	2.00	1			1		
Non-users	37,210	243,515	399	1.64	0.77	0.70–0.85	<0.001	0.76	0.69–0.84	<0.001
Sensitivity analysis 3 ^§^										
SSRIs users	115,851	804,265	1610	2.00	1			1		
Non-users	38,617	253,017	417	1.65	0.77	0.70–0.85	<0.001	0.77	0.70–0.85	<0.001
Sensitivity analysis 4^||^										
Non-users	24,930	235,566	415	1.76	1			1		
TCA users	8310	74,980	118	1.57	0.85	0.71–1.02	0.088	1.02	0.76–1.39	0.875

* Adjusted for age at index, sex, education, birth country, family history of CRC, history of inflammatory bowel disease, COPD, obesity, history of colonoscopy, use of aspirin, use of statin, use of metformin, CCI. ^†^ Sensitivity analysis 1: Use of SSRIs and the risk of CRC among individuals with a family history of CRC lagging one year after the first administration of SSRIs. ^‡^ Sensitivity analysis 2: Use of SSRIs and the risk of CRC among individuals with a family history of CRC after excluding individuals with a benign colorectal tumor. ^§^ Sensitivity analysis 3: Use of SSRIs and the risk of CRC among individuals with a family history of CRC. Both death and diagnosis of other cancers were modeled as competing events. ^||^ Sensitivity analysis 4: Use of TCA and the risk of CRC among individuals with a family history of CRC.

**Table 5 cancers-14-05905-t005:** Sensitivity analysis 5. Mendelian randomization analysis for the causal effect between depression and colorectal cancer.

Method	Beta	SE	Causal Effect (95%CI)	*p* Value
MR Egger	5.345 × 10^−3^	7.102 × 10^−3^	1.005 (0.991–1.019)	0.449
Weighted median	1.106 × 10^−3^	6.352 × 10^−3^	1.001 (0.989–1.013)	0.862
Inverse variance weighted	3.836 × 10^−3^	4.043 × 10^−3^	1.004 (0.996–1.012)	0.343

## Data Availability

The data based on the Swedish register are not publicly available due to Swedish law and protecting patients’ privacy, and the combined set of data used for the analysis presented in this study can only be made available from the appropriate Swedish authorities (the Swedish National Board of Health and Welfare (https://www.socialstyrelsen.se/en, accessed on 4 May 2021) and Statistics Sweden (https://www.scb.se/en, accessed on 7 September, 2021),) and Statistics Sweden (https://www.scb.se/en, accessed on 7 September, 2021), for researchers who meet the criteria for access to confidential.

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
