# Peer review of "Use of Selective Serotonin Reuptake Inhibitors Is Associated with a Lower Risk of Colorectal Cancer among People with Family History"

_cancers, 2022, doi:10.3390/cancers14235905_

Round 1

Reviewer 1 Report

This paper utilized Swedish national registry data to explore the potential protective effect of SSRIs against colorectal cancer among a specific high-risk population. Overall, it is a well-designed and written paper. The sensitivity analysis provides additional value when interpreting the study result.

I only noticed 3 issue that need to be addressed. 

1. Line 113-114 stated "This study excluded patients with a follow-up period of fewer than three months....". While in Figure 1, follow-up time less than one year were excluded. Please double check.

2. Line 170-174, the description is opposite from table 1. E.g., it says "SSRIs users had a slightly higher proportion of highest education level...". But in table 1, SSRIs vs. Non-users among >=12 years of education is 31.0% vs. 31.6%.

3. Line 185, "...due to the very limited number of cases" indicates it's a confirmed and only reason no significant difference observed in fluoxetine, paroxetine, or escitalopram. I suggest adding words like "likely due to" to be more accurate. 

Author Response

1. Line 113-114 stated "This study excluded patients with a follow-up period of fewer than three months....". While in Figure 1, follow-up time less than one year were excluded. Please double check.

Response: Thank you for your reminder. We double-checked our data, and revised the information in Figure 1 into “follow-up time fewer than three months”

2. Line 170-174, the description is opposite from table 1. E.g., it says "SSRIs users had a slightly higher proportion of highest education level...". But in table 1, SSRIs vs. Non-users among >=12 years of education is 31.0% vs. 31.6%.

Response: Thank you for pointing out the mistake, we revised the description in the results section. Please see more details in lines 168-174.

3. Line 185, "...due to the very limited number of cases" indicates it's a confirmed and only reason no significant difference observed in fluoxetine, paroxetine, or escitalopram. I suggest adding words like "likely due to" to be more accurate. 

Response: We appreciate your suggestion and revised the text accordingly.

Reviewer 2 Report

Zhang and colleagues examined the effects of selective serotonin reuptake inhibitors on the risk of colorectal cancer using a nationwide cohort study. Overall, the study was well designed and examined the effects of SSRIs on colorectal cancer well. It would be more beneficial for the paper, if authors examined deeper into specific colorectal subtypes as well as treatments that subjects are undertaking. 

According to figure 1. Looks like a significantly more portion of people who took SSRI died compared to controls. Can you provide the analysis in the manuscript and explain in the discussion why this was the case. Does this higher number of those who took SSRI died compared to controls directly involved in SSRI intake?

Please examine the colorectal cancer subtypes (MSI and MSS). Does colorectal type affect the effectiveness of SSRI on colorectal cancer risk?

Does SSRI intake affect chemotherapy treatment of colorectal cancer patients? (eg those who are undertaking chemotherapy in stage III and IV colorectal cancer?

Did SSRI interacted with other drugs? Such as Aspirin, metformin and statin? 

Please provide examples and comparison of SSRI like drugs in the discussion. How the performance of SSRI compares to other similar drugd?

Author Response

According to figure 1. Looks like a significantly more portion of people who took SSRI died compared to controls. Can you provide the analysis in the manuscript and explain in the discussion why this was the case. Does this higher number of those who took SSRI died compared to controls directly involved in SSRI intake?

Response: Thank you for raising this question. The reason for the high proportion of death in the SSRI users’ group is that most of them have depression or anxiety which were risk factors for suicide. By estimate, the suicide rate was 0.04% per year in depression patients whereas 0.0036 to 0.037% in the general population [1]. Besides, our study population was the elder population, and the mortality rates are comparatively high. In consideration of this issue, we computed death as a competing event to control the competing risk of death. We added the information mentioned above in the method section, please see more details in lines 130-135.

Please examine the colorectal cancer subtypes (MSI and MSS). Does colorectal type affect the effectiveness of SSRI on colorectal cancer risk?

Response: We agree that there might be heterogeneity in different CRC subtypes. Unfortunately, we lack this information in our registers, so we are not able to perform this analysis. We also discussed this limitation in the manuscript.

Does SSRI intake affect chemotherapy treatment of colorectal cancer patients? (eg those who are undertaking chemotherapy in stage III and IV colorectal cancer? 

Response: SSRI might be a potential agent to treat the adverse effect of chemotherapy. Oxaliplatin is a commonly used platinum-based chemotherapy drug for colorectal cancer. SSRI was reported to inhibit oxaliplatin-induced peripheral neurotoxicity in a dose-dependently way [2]. We added the information mentioned above in the discussion section, please see more details in lines 295-298.

Did SSRI interact with other drugs? Such as Aspirin, metformin and statin? 

Response: One previous publication from our research group suggested that there might be an interaction between aspirin and SSRIs against CRC [3]. We additionally adjusted metformin and statin in considering that there might be potential unknown drug-drug interaction. Besides, use of those medications also represents a lifestyle choice or health behavior. So we adjusted the use of aspirin, metformin, and statin in the multivariate model.

Please provide examples and comparison of SSRI like drugs in the discussion. How the performance of SSRI compares to other similar drugs?

Response: We did a sensitivity analysis to evaluate possible indication bias associated with SSRI use by investigating the relationship between tricyclic antidepressant (TCA) use and CRC risk. In Sweden, TCA is also an extensively prescribed agent for the management of depression. The result showed that TCA use was not associated with CRC risk, with an adjusted HR of 1.02 (95% CI, 0.76–1.39). Compared with TCA, SSRIs have a stronger protective effect against CRC, indicating that our result was not confounded by indication bias. Please see more details in Table 4, as well as in the discussion section 318-322.

Reference

  1. Miller, J.N.; Black, D.W. Bipolar Disorder and Suicide: a Review. Curr Psychiatry Rep 2020, 22, 6, doi:10.1007/s11920-020-1130-0.
  2. Kang, L.; Tian, Y.; Xu, S.; Chen, H. Oxaliplatin-induced peripheral neuropathy: clinical features, mechanisms, prevention and treatment. J Neurol 2021, 268, 3269-3282, doi:10.1007/s00415-020-09942-w.
  3. Zhang, N.; Sundquist, J.; Sundquist, K.; Zhang, Z.G.; Ji, J. Combined Use of Aspirin and Selective Serotonin Reuptake Inhibitors Is Associated With Lower Risk of Colorectal Cancer: A Nested Case-Control Study. Am J Gastroenterol 2021, 116, 1313-1321, doi:10.14309/ajg.0000000000001192.